# CLINICAL RISK: WAVELET RECONSTRUCTION NETWORKS FOR MARKED POINT PROCESSES

## ABSTRACT

Timestamped sequences of events, pervasive in domains with data logs, *e.g*, health records, are often modeled as point processes with rate functions over time. Leading classical methods for risk scores such as Cox and Hawkes processes use such data but make strong assumptions about the shape and form of multivariate influences, resulting in time-to-event distributions irreflective of many real world processes. Recent methods in point processes and recurrent neural networks capably model rate functions but may be complex and difficult to interrogate. Our work develops a high-performing, interrogable model. We introduce wavelet reconstruction networks, a multivariate point process with a sparse wavelet reconstruction kernel to model rate functions from marked, timestamped data. We show they achieve improved performance and interrogability over baselines in forecasting complications and scheduled care visits in patients with diabetes.

## 1 INTRODUCTION

Clinical risk scores are commonly used analytic devices in health care. There are risk scores for predicting strep throat from sore throats (Centor et al., 1981), mortality from vital signs (Gardner-Thorpe et al., 2006), heart attacks from routine clinic visits (D'Agostino et al., 2008), and many more. Policy is implemented around these risk scores, from rates of reimbursement to physician compensation (Asch et al., 2015). When used for early warning, risk scores have been associated with reduced mortality (Seymour et al., 2017).

Underlying these approaches is the formulation of risk over time given some set of features. For example, in Cox models, a study time $t_0 = 0$ is defined and the risk model is $\lambda(t; t', x)$, where $x$ are defined by features timestamped before time $t' = t_0$, *i.e.*, a time invariant model. In Hawkes processes, the risk model is also $\lambda(t; t', x)$ but where $x$ contains all history up to time $t' = t$, *i.e.*, a nowcasting model. Cox and Hawkes models can be limited by their assumptions, which often are inappropriate for the health care setting, and include, for example, the assumptions of proportional hazards and summation over kernel activations. We formulate a point process model to address these limitations and develop multi-forecasting, the forecasting task across the two dimensions of time: $t'$ and $t$.

To motivate our specific formulation, consider the limitations of the Hawkes process in health care. First, the Hawkes process encodes an additive relationship of change in rate from recurring precursors, *i.e.*, burstiness, whereas in health care, the repeated measurement of an event beyond the first, say of glucose, might be irrelevant. Second, clinical event timing may be routine, scheduled, or emergent, which suggests that kernel learning will improve model performance because changes in the rate may be time-dependent and not immediate. Third, clinical event processes are marked, with marks that could be categorical, real, or null values: *e.g.*, bacterial culture: staphylococcus aureus, glucose: 200, and ketoacidosis: NA

Our model addresses each of these limitations. To address the first limitation, summation, we adopt a reduction layer where we allow for reductions other than "sum" of kernel contributions from recurring events. To address the second and third limitations, non-specific timing and lack of marks, we propose a kernel learning method over one dimension (time) and two dimensions (time-value) using wavelet reconstructions. The motivation for wavelets is illustrated in Figure 1, where a discrete wavelet reconstruction encodes the relationship of time-delayed events identified through cross-correlation. While cross-correlations capture all relative timings, many may be spurious or coincidental. The

wavelet representation of relative timing instead is learned through likelihood optimization. To capture the effect of the value distribution of marks on events, two-dimensional wavelet reconstructions are value indexed, producing a one-dimensional reconstruction that is passed forward to the reduction layer. We show that we can encode the wavelets, the relative-to-absolute mapping, and the reduction step on a computation graph to conduct learning.

We apply our model first in simulations of heart attacks and scheduled hemoglobin A1c checks. We then evaluate the model in a real cohort to forecast complications and adherence to clinical participation in patients with diabetes. Empirically, we show that our model has improved performance over baselines in single forecasts and explore its performance in multi-forecasts in terms of prediction and representation.

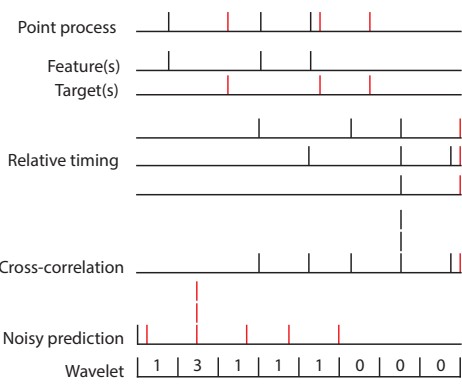

Figure 1: Illustrative 1-d cross-correlation motivating the discrete wavelet reconstruction kernel for relative time dependencies.

Our results provide benefit for diabetes risk assessment. Whereas the central focus of the diabetes adherence literature is on adherence to medication and therapeutic regimen (García-Pérez et al., 2013; Edelman and Polonsky, 2017), our method provides a lens to investigate adherence to continued diabetes care participation, which is important because, for one, participation is associated with better diabetes control (Schectman et al., 2008). Additionally, forecasts of complications and adherence to participation, at different times in the care process and at different forecast distances, are central concerns in clinical decision-making. For example, many diabetes medications require regular clinical monitoring, and the risk of complications and noncompliance to clinical participation both affect regimen choice. By focusing on these outcomes, our forecasting tools provide relevant information for the clinical decision-making process.

**Related work.** Both neural networks and Hawkes process variants are used for rate modeling in health care: a few recent examples include Choi et al. (2015), Du et al. (2016), Alaa et al. (2017), and Bao et al. (2017). To effectively model the hazard two key properties are used: (1) the ability of Hawkes processes to capture relative timing of interdependent events, and (2) the flexible functional forms of neural networks that are able to capture relative timings, albeit somewhat opaquely. The closest work is likely that of Bao et al. (2017), where the authors adopt dyadic influence functions. However in that work marks are not used and the dyads selected are a subset of the Haar wavelet basis. Outside of health applications, related literature includes Hawkes kernel learning, *e.g.* Zhou et al. (2013); Linderman and Adams (2014); Lee et al. (2016). Our method follows these approaches, instead using a wavelet representation across value and time to capture long-range dependencies. Compared to Mei and Eisner (2017) that generalizes the Hawkes process with neural networks, our method (1) allows events to have marks, (2) enables forecasting and multi-forecasting (not just nowcasting), and (3) can be seen to have a generalized linear form.

Our method of mapping relative timing hazard components onto absolute time possesses the advantages of the Hawkes approach and adopts a simple but well-performing neural network architecture. To achieve the mapping, a stepwise hazard approximation is made, as is done in Jing and Smola (2017) and Weiss (2017), however, instead of LSTMs and forests that have challenging interpretations, our method remains interpretable for small data sets, where exploration with visualization similar to that of Caruana et al. (2015) can be performed. Our method uses wavelets to represent event contributions, and several survival analysis approaches have also adopted them in univariate models, *e.g.*, in Antoniadis et al. (1994) and Brillinger (1997). Outside of survival analysis and point processes, wavelet-inspired neural networks have seen success, with Wave-net using wavelets to classify time series (Bakshi and Stephanopoulos, 1993), and Wavenet adopting a multi-layer hidden neural architecture to connect distant time steps (Van Den Oord et al., 2016).

**Contributions.** The contributions of this work are as follows: our work generalizes multivariate Hawkes processes to allow for non-additive event rate relationships. Like other works, *e.g.* Linderman and Adams (2014), our work learns the kernel function that relates multivariate event histories to the

rate. However, in our work we use wavelets as the kernel, akin to a multivariate development from Brillinger (1997). We leverage the scaling property of wavelets to formulate a regularization that balances spatiotemporal generalizability with deterministic or near-deterministic event timing. Unlike many sequence models, *e.g.* Hochreiter and Schmidhuber (1997), which are affected strongly by choice of time step, our work adopts an absolute and relative time frame, and therefore the granularity of the absolute time domain need not be determined a priori. Additionally, unlike some point process formulations, our work models marks that are 1-d (event times) and 2-d (event times and their category or real value). Our work is explicit about types of forecasting tasks, and the methods are adapted for several purposes: forecast performance, interrogability, and representation. We show that our method performs as well as or better than comparison algorithms in predicting complications and forecasting adherence.

## 2 BACKGROUND

Let $E$ be the set of events with target event $y \in E$ the event we want to forecast. Associated with each event $e$ is a value $v \in V$. An example consists of a sequence of (time, event, value) tuples and a period of interest for forecasting. For the $n$-th example, $n \in \{1 \ldots N\}$, define $T_n$ as the number of tuples. Then the sequence can be written as $(t_{in}, e_{in}, v_{in})$ for $i \in \{1 \ldots T_n\}$, with the period of interest denoted as $\tau_{ny}$.

Let $\lambda_y(t)$ be the rate functions of interest, dropping the subscript $n$ for ease of notation. The multivariate Hawkes process can then be written as follows:

$$\lambda_y(t) = \lambda_0(t) + \sum_{e=1}^{|E|} \beta_e \sum_{i=1}^{T} g_e(t - t_i) \mathbb{1}(t_i < t, e_i = e)$$

where $\lambda_0(t)$ is a baseline population rate function, $g_e(\cdot)$ is a kernel function for event $e$ relating its effect on the rate of $y$, $\beta_e$ are event-specific parameters, and $\mathbb{1}(\cdot)$ is the indicator function. Typically $g_e(\cdot)$ is an event-specific exponential decay function with a learnable decay parameter. Self-exciting processes are defined by $g_y(\cdot) > 0$, bursty processes by $g_e(\cdot) > 0$, and inhibitory processes by $g_e(\cdot) < 0$. A few recent variations include Linderman and Adams (2014) where $g_e$ is a Bayesian graph kernel and Xu et al. (2017) where $g_e$ is an infectivity function and triggering kernel product.

Given $\lambda_{ny}(t)$, the log likelihood of the data is:

$$\text{LL}(X|\theta) = \sum_{n=1}^{N} \Big( \sum_{i=1}^{T_{ny}} \log \lambda_{ny}(t_{iny}) + \int_{\tau_{ny}} \lambda_{ny}(t)dt \Big) \tag{1}$$

The form of the Hawkes process is limiting, however, because (1) the effect of $g_e(\cdot)$ decays over time, (2) the effect over $g_e(\cdot)$ is additive, (3) the value associated with each event is not considered, and (4) the time restriction in the indicator function implies nowcasting ($\mathbb{1}(t_i < t)$) not forecasting ($\mathbb{1}(t_i < t - c)$ for some $c > 0$). Making modifications to achieve these characteristics is desirable to effectively model many real-world processes. For example, a patient with new-onset diabetes schedules an appointment with a typical gap interval of 3 months, and the presence of 1 or 10 elevated readings may not affect the timing of the scheduled appointment. The former suggests the utility of kernel learning, and the latter suggests summation over $g_e$ is not the appropriate reduction.

The proposed method addresses these concerns. In particular, we adopt discrete wavelet reconstructions, which both allows kernel learning and the use of marks. Additionally, by representing the point process as a neural network, we are able to (1) use maximization alongside summation in a reduction layer (the formulation enables specification of any number of reductions), and (2) conduct time-dependent censoring. We formalize the model below.

## 3 WAVELET RECONSTRUCTION NETWORKS

We now define wavelet reconstruction networks (WRNs) shown in Figure 2. We specify the form of the rate function, define our kernel function, and impose restrictions on the kernel function for forecasting and multi-forecasting.

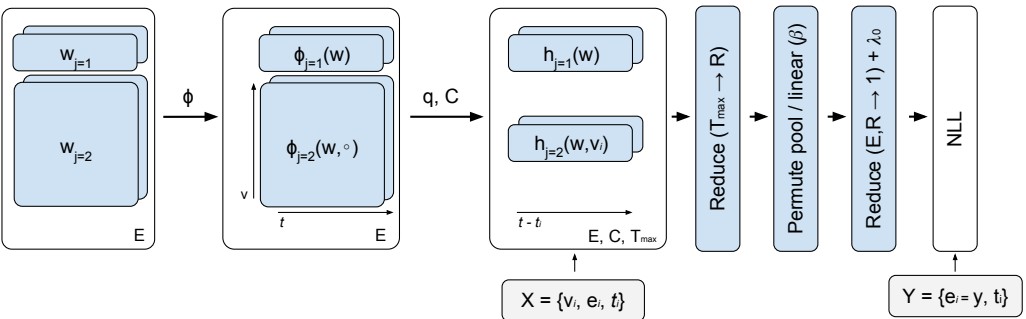

Figure 2: Wavelet reconstruction network architecture

Let $h_{ej}(t; t_i, \tau_d)$ be a piecewise-constant kernel function for event $e$ on absolute time intervals $\tau_d$ with index set $J = \{1, 2\}$, where $j = 1$ indicates a time kernel and $j = 2$ indicate a time-value kernel. Let $R$ be a set of reduce functions, which for our model we set $R = \{\text{sum}, \text{max}\}$. Then $r_{i=\{1,...,T\}}^{\tau_y}$ indicates the reduction over $T$ functions over the interval $\tau_y$. We propose the following rate function:

$$\lambda_y(t) = \lambda_0(t) + \sum_{e=1}^{|E|} \sum_{r,j}^{R,J} \beta_{erj} r_{i=\{1,...,T\}}^{\tau_y} \big( h_{ej}(t; t_i, \tau_d) \mathbb{1}(e_i = e) \big). \tag{2}$$

Note that the kernel function $h_{ej}$ is on absolute time, whereas $g_e$ is defined on relative time in the previous section. This is done to specify the translation of the wavelet reconstructions on discrete relative-time intervals onto discrete absolute-time intervals, which requires additional treatment to prevent causal leakage. We define the kernel function on relative time below, followed by the definition of $h_{ej}$ using the translation function.

**Discrete wavelet reconstruction kernel.** Recall that the discrete wavelet transform (DWT) is an invertible transform of a signal between a time space and a time-frequency space used in multi-resolution analysis and signal compression (Mallat, 1989). Here we use the inverse DWT to encode the parameters in the time-frequency space to reconstruct the signal as the kernel function as follows.

For each event, we use one- to two- dimensional wavelets, with $j = 1$ referring to time reconstructions, and $j = 2$ referring to time and event value reconstructions. We use discrete wavelets of size $(\alpha)$ and $(\alpha, |S|)$, with $\alpha$ time intervals on the interval $[0, \max_n(\max_{t_{iny}} t_{iny} - \min_{t_{ine}} t_{ine})]$ on the time dimension, and with $s \in S$ disjoint value intervals in $[min(v), max(v)]$ on the value dimension. For point events, we use the notation $s \in S$ where $|S| = 1$, and categorical events are treated as separate point events. Note that $j = 1$ reconstructions are the temporal analogues of missingness indicators.

Let parameters $w_{ej}$ be the wavelet coefficient tensors for event $e$ and wavelet reconstruction dimension $j$, and let $g_{ejs}$ be the kernel function on relative time for event $e$ and interval $s$, with $g_{ejs}(t - t_i) = 0$ for $t - t_i < 0$. Define the set of wavelet reconstruction functionals by: $\Phi = \{\phi_{ej} : (w_{ej}, v_e) \mapsto g_{ejs}\}$. Conceptually, given event value $v_e$, the wavelet reconstruction functional $\phi_{ej}$ reconstructs the signal from $w_{ej}$ and indexes the value dimension with $v_e$, producing function $g_{ejs}$, a function with inputs of relative time.

**Relative- to absolute-time transformations.** To relate the absolute-time kernel $h_{ej}$ with the relative-time discrete kernel $g_{ejs}$, we define the following causally-protective translation function. Let $\tau_d$ denote disjoint caglad intervals that comprise $\tau_y$ the target interval in absolute time, and let $\tau_e$ be the relative-time intervals of the wavelet reconstruction. We denote lower and upper endpoints with $\lfloor \cdot \rfloor$ and $\lceil \cdot \rceil$. Then, for event time $t_i$, the absolute wavelet reconstruction intervals have endpoints $\lfloor \tau_{ie} \rfloor = \lfloor \tau_e \rfloor + t_i$ and $\lceil \tau_{ie} \rceil = \lceil \tau_e \rceil + t_i$. The transformation $q$ is given by:

$$h_{ej}(t; t_i, \tau_d) = q\big(g_{ejs}(t - t_i), \tau_d\big) = \frac{\sum_{\tau_{ie} \wedge \tau_d \neq \varnothing, \lfloor \tau_{ie} \rfloor \geq \lceil \tau_d \rceil} (\max(\lfloor \tau_d \rfloor, \lceil \tau_{ie} \rceil) - \lfloor \tau_{ie} \rfloor) g_{ejs}(t - t_i)}{\lceil \tau_d \rceil - \lfloor \tau_d \rfloor},$$

where $\wedge$ denotes interval intersection. The second condition for inclusion in summation, $\lfloor \tau_{ie} \rfloor \geq \lceil \tau_d \rceil$, prevents causal leakage by ignoring intervals $\tau_{ie}$ that affect an interval $\tau_d$ that both precedes and intersects it.

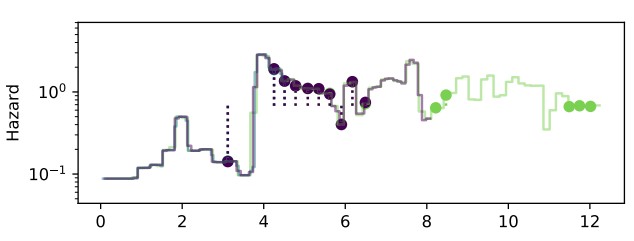 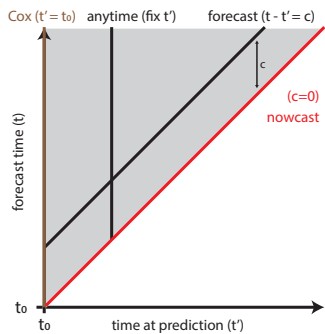

Figure 3: Left: three absolute time hazards (green, blue, and purple) for one trajectory with different steps sizes and censor times. Right: forecasting tasks. The vertical distance from the line $t' = t$ is censor distance $c$. Common loss definitions are along solid lines.

One advantage of our relative-time specification is that the granularity over absolute time can be adjusted with small effect on the hazard. By comparison, RNNs and their analogues need to be retrained if there are changes to the time-step specification. Figure 3 (left) illustrates this with an absolute-time hazard function of a single trajectory using different absolute time steps and applying censorship at different times, resulting in absolute-time hazards similar but not identical. Sharp discontinuities near target event times can result in relatively large likelihood differences, and the ability to choose the absolute time granularity $\tau_d$ post-training facilitates hazard recovery as needed. Figure 3 (left) also acts as a causality leakage check by demonstrating that the hazard is unaffected by the presence or censorship of future events.

**Forecasting.** Thus far we have specified $h_{ej}$ as a nowcasting kernel because $g_{ejs}$ is zero and non-contributory when $t - t_i < 0$ for all $e$ and $s$. For forecasting, we incorporate time-dependent censoring with functional $C$, the Hadamard censor, and hyperparameter $c$ the censoring distance, to prevent recent events, *i.e.*, risk modifiers, from affecting the rate. Let $C(c)(t; t_i)$ equal 1 if $t - t_i > c$ and 0 otherwise, and let $\psi_{ejc}(w_{ej}, v_e) = \psi_{ej}(c)(w_{ej}, v_e) = C(c) \circ \phi_{ej}(w_{ej}, v_e)$ where $\circ$ is the Hadamard product. We can specify the forecasting rate function analogous to Equation 2 as follows:

$$\lambda_{yc}(t) = \lambda_0(t) + \sum_{e=1}^{|E|} \sum_{r,j}^{R,J} \beta_{erj} r_{i=\{1,\dots,T\}} \big( q(\psi_{ejc}(w_{ej}, v_i); \tau_d) \mathbb{1}(e_i = e) \big) \tag{3}$$

For multi-forecasting, we choose a vector of desired relative forecast times $\{c\}$ and maximize the average log-likelihood over all $c$. This may be distinguished from training separate forecasting models because the parameters of the model are tied. Figure 3 (right) contrasts the forecasting tasks. Compared to single forecasts along any given line, multi-forecasting captures more of the valid forecast region (shaded gray) that is relevant for clinical decision making.

**Learning.** The parameters of the model are $\Theta = \{w_{ej}, \beta_{erj}\}$. Because the system may be overdetermined, we add regularization terms. The first is $\gamma_\beta \sum_e \sum_{rj} ||\beta_{erj}||_1$ akin to the LASSO (elastic-net regularization is equally straightforward). The second is the regularizer $\gamma_w \sum_{ej} ||u(w_{ej})||_1$ akin to sparse shrinkage on the wavelet tensor with a choice for $u$.

We define $u(w_{ej}) = \bigotimes_{k \in \{1,\dots,j\}} 2^{l_k/2} \circ w_{ej}$, where $l_k$ is the wavelet scale parameter of the $k$-th dimension. The idea is that regularization on wavelets for point events corresponds to smoothing a function of Dirac deltas over time, and we want the log loss effect of a Dirac delta (an element of the first term in Equation 1) to be in proportion to the activation of the unnormalized wavelet basis function so that the data drive the choice of smoothness. To do so, the regularization must be in proportion to $\bigotimes_{k \in \{1,\dots,j\}} 2^{l_k/2}$. An example is the orthonormal two-level Haar wavelet, where the orthonormal transformation matrix is written as the Hadamard product of expanded, exponentiated scale parameters and unnormalized basis functions, where $\mathbf{1}$ is a column vector of ones:

$$\left( \begin{bmatrix} 2^{-1} \\ 2^{-1} \\ 2^{-1/2} \\ 2^{-1/2} \end{bmatrix} \mathbf{1}^\intercal \right) \circ \begin{bmatrix} 1 & 1 & 1 & 1 \\ 1 & 1 & -1 & -1 \\ 1 & -1 & 0 & 0 \\ 0 & 0 & 1 & -1 \end{bmatrix}.$$

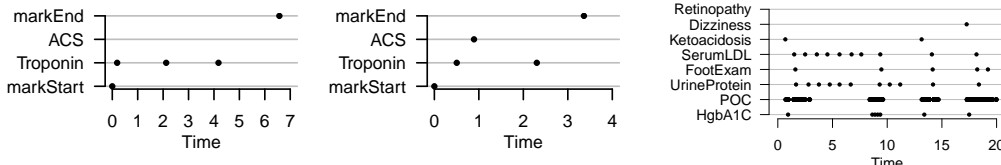

Figure 4: Examples: acute coronary syndrome–angina (left), ACS (middle); diabetes simulation (right).

**Improving prediction.** The formulation in Equation 3 can be seen as a generalized linear model of censored, add-/max- reductions of wavelet reconstructions. While the generalized linear form lends itself to interpretation, we consider whether non-linearities will further improve predictive performance. Unlike images, where local pixels are spatially related, any given ordering of clinical event types may not capture valuable event relationships. Therefore we introduce permute-and-pool layers (WRN-PPL) that randomly permute event ordering within time step, randomly select sign, perform max-pool, and project linearly to the next layer. In place of the double summation in Equation 3, we apply a random sign ($\{-1,1\}$) Hadamard tensor $Z$ and pass the result to $P$ parallel permutation layers with max pools of size $\min(2^p, |E||J||R|)$ for $p = 0$ to $P - 1$. The outputs of the max pool are then linearly combined and output to the next layer.

## 4 EXPERIMENTS

We conduct tests in two simulations and on one real health care data set. For direct comparison, we evaluate performance against methods in the nowcasting framework using negative log likelihood and ranking measures. Then we explore WRN and WRNPPL in forecasting and multi-forecasting.

**Setup.** We divide the data into a train, tune, and held out test set. Model development is performed on train and tune sets with parameters determined by early stopping. Models are then evaluated on the held out test set. We use the Goodman-Kruskal $\gamma$ statistic as a measure of concordance among non-tied pairs: (agree - disagree - prior)/(agree + disagree + prior), where the prior penalizes algorithms providing identical predictions. Agreement occurs when the predicted average rate and the empirical average rate are both greater in one example compared to the other, ties when either is the same, and disagreement otherwise. A crude interpretation of $\gamma$ is the difference $(1 - \gamma)/2$ gives the width of a band corresponding to random guessing of correct ordering, and correctly predicting all other pairs. Further details of parameter setting, *e.g.* preprocessing, bin widths, optimizer settings, are in the Appendix. The code is written in PyTorch 0.4.1 and will be released upon publication.

**Comparison methods.** We compare wavelet reconstruction networks (WRNs) with homogeneous Poisson processes, time-invariant and nowcasting Fourier basis functions, multivariate Hawkes processes, and two long short-term memory (LSTM) networks. Briefly, the Fourier methods are given by $f(t|t_0, x) = \sum_k \sum_l w_{kl} sin((2\pi l/\tau)(t - t_0)) + v_{kl} cos((2\pi l/\tau)(t - t_0))x_k$, where $x$ corresponds to features, *i.e.*, event by value interval occurrences, at or before time 0 ($t_0 = 0$, time-invariant) and at $t_0 = t$ (nowcasting), $k$ indexes the event by value interval by time step, and $l$ indexes the basis function component. L2 regularization are applied to $w_{kl}$ and $v_{kl}$. Then the rate function is defined as $\lambda(\cdot) = w_0 + f(\cdot)^2$. For nowcasting, the Fourier method is given features from 16 previous time steps, and since in nowcasting $t - t_0 = 0$, the formula reduces to a generalized linear model of $v_{kl}x_k$ terms. The multivariate Hawkes process we use includes a kernel with event-specific exponential decay parameter $\gamma_e > 0$: *i.e*, $g_e(t - t_i) = e^{-\gamma_e(t - t_i)}$. We use a learnable constant baseline rate $\lambda_0$. We learn $\beta_e$ without constraint, rather than $\beta_e \geq 0$ or $\beta_e \leq 0$ of Hawkes and inhibitory processes respectively. We apply a positivity constraint to ensure the rate is non-negative.

The first LSTM method is a variant of the multi-task healthcare LSTM from Lipton et al. (2016) where the preprocessing involves zero- or last-value carry forward- imputation, mean-reducing, and adding missing indicators. Because our task is nowcasting not multi-label classification, we modify the loss function accordingly. The second LSTM is a WRN preprocessing LSTM analogue. The LSTM includes a linear-embedded input ($i \times h$), two LSTM hidden layers ($h \times h$), and output to a rectified linear layer ($h \times 1$) where $h$ is the hidden unit width. For each model, the output is a hazard per time step $\lambda_{ny}$, and the loss is the point process log likelihood in Equation 1.

Table 1: Negative log likelihood on the held out test set. Asterisk (*) denotes simulation; KNR: {ketoacidosis, neuropathy, retinopathy}; H. Poission: homogeneous Poisson; Time-invariant: Fourier prediction at time $t = 0$; Nowcast: 16-step history as nowcast features. LSTM[1]: Lipton et al. (2016), best of zero-imputation or last-value carry forward, with missingness indicators; LSTM[2]: LSTM with WRN preprocessing; WRN: wavelet reconstruction network; PPL: with permute pool layer. Best performer in bold.

| Dataset | Method (NLL) | | | | | | | |
|---|---|---|---|---|---|---|---|---|
| | H. Poisson | Time-invariant | Nowcast | Hawkes | LSTM[1] | LSTM[2] | WRN | WRN-PPL |
| ACS* | 0.44 | 0.43 | 0.36 | 0.39 | 0.21 | **0.13** | 0.23 | 0.15 |
| A1c* | 18.54 | 19.20 | 13.56 | 3.87 | 11.80 | 4.10 | 3.93 | **3.78** |
| A1c | 2.86 | 2.76 | 2.52 | 1.67 | 1.15 | 1.29 | 1.23 | **1.13** |
| KNR | 0.75 | 0.71 | 0.58 | 0.31 | 0.46 | 0.35 | **0.24** | 0.26 |

Table 2: Goodman-Kruskal $\gamma$, a measure of concordance, on the held out test set. Asterisk (*) denotes simulation.

| Dataset | Method (Goodman-Kruskal $\gamma$) | | | | | | | |
|---|---|---|---|---|---|---|---|---|
| | H. Poisson | Time-invariant | Nowcast | Hawkes | LSTM[1] | LSTM[2] | WRN | WRN-PPL |
| ACS* | -1.00 | -0.81 | 0.08 | -0.97 | **0.98** | 0.91 | 0.80 | 0.85 |
| A1c* | -1.00 | -0.02 | 0.81 | 0.64 | 0.84 | **0.85** | 0.78 | 0.77 |
| A1c | -1.00 | 0.23 | 0.71 | 0.79 | 0.83 | 0.87 | 0.93 | **0.93** |
| KNR | -1.00 | 0.25 | 0.81 | 0.91 | 0.84 | 0.91 | 0.95 | **0.98** |

**Simulations.** The first simulation is of heart attack diagnoses denoted by acute coronary syndrome (ACS). In this simulation, it is the elevation in value of troponin, a heart enzyme measurement, outside the normal range (less than 0.01 ng/mL) that indicates ACS will occur in the next time unit uniformly at random. Figure 4 (left, middle) illustrates two trajectories; note both the mark and the timing are important for ACS determination. The second simulation is of diabetes care: patients with diabetes undergo semi-regular appointments, *e.g.*, annual eye and foot exams, quarterly hemoglobin A1c measurements, and pre- and post-prandial glucose measurements. These patients are often non-adherent with worsening adherence as a function of increasing time from adverse events. Figure 4 (right) illustrates the timings of an example trajectory.

**Diabetes visits.** We partnered with a regional health care system to investigate the risk of adverse outcomes of diabetes and adherence to the care those patients received. From the regional cohort followed from 2010 to 2017, we selected those at risk of diabetes as defined by an outpatient measurement of hemoglobin A1c or glucose, or a diagnosis of hyperglycemia. Among those, we excluded any individuals without at least two clinic encounters more than six months apart. We additionally applied a censor date at the time of the last clinical event before a 30-month gap in care, where there is uncertainty that the patient is lost to follow-up or is receiving care outside of network.

Application of the inclusion and exclusion criteria resulted in 798,818 timestamped events in a study population of 4,732 individuals each representing a single example. We divided the population into thirds: {train, tune, test} sets. We focused on two outcomes: (1) hemoglobin A1c measurements, as a proxy for scheduled diabetes care, and (2) a combined outcome of {ketoacidosis, neuropathy, retinopathy} as defined by ICD 9 and ICD 10 codes. Features included were extracted with string matching on event descriptions of events documented as putative risk factors in clinical guidelines from the ADA, AHA, and UpToDate, and included events from demographics, medications, encounters, laboratory, diagnosis, and procedures tables. The extraction resulted in 575 features. Hemoglobin A1c was measured at least once in 820 individuals (21%), and an adverse event occurred at least once in 137 individuals (3%). Additional details are given in the Appendix.

## 5 RESULTS

**Nowcasting.** Table 1 reports the negative log likelihoods for the experiments on the held out test sets. Overall, the proposed wavelet reconstruction network WRN-PPL outperformed the other algorithms. The WRN-PPL method excelled particularly in tasks with many target occurrences (A1c* and A1c experiments) and performed near to the best in rare occurrence data (ACS* and KNR). The WRN

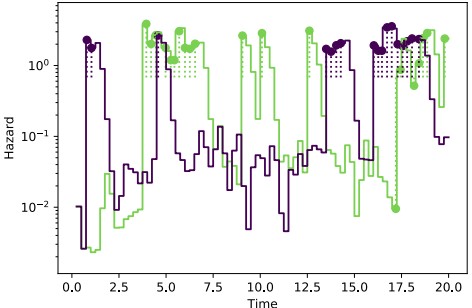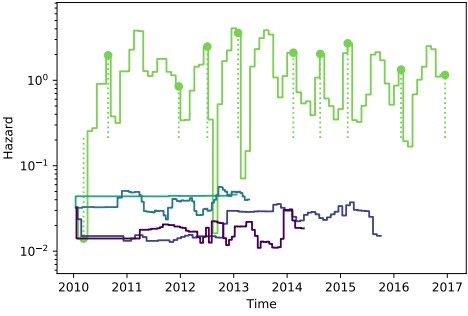

Figure 5: Left: hazards and hemoglobin A1c events for A1C* simulation. Right: hazards and hemoglobin A1c events for five random, test set patients in real cohort.

method outperformed the WRN-PPL method at the KNR task, which could be due to the relatively low rate of target events (0.025 per year) where complexity may lead to overfitting; however the difference in negative log likelihood is small. The WRN and WRN-PPL $\gamma$ statistics showed effective risk stratification of individuals with diabetes. A $\gamma$ of 0.98 corresponds to a band of 0.01 under the crude interpretation and suggests strong overall risk stratification.

At the other end, the time-invariant method performed marginally better than homogeneous Poisson. The nowcast method performance shows the utility of incorporating time-varying information but substantially underperformed compared to the other methods. The Hawkes process also lacked performance with comparable NLL only in the A1c* data set. LSTM[1] had good ranking characteristics on the simulations but mediocre predictive performance with notably poor performance on the simulation A1c*, where the timing of events, not their value distribution, determines the ground truth rates. LSTM[2] mostly outperformed LSTM[1], suggesting the usefulness of WRN preprocessing, but mostly underperformed against WRN-PPL.

Figure 5 shows the hemoglobin A1c predicted hazards profile for random test set patients using WRN-PPL. The step function represents predicted hazard over time, points indicate true event times and dotted lines show the difference from the baseline (homogeneous Poisson) rate. The WRN-PPL algorithm makes predictions that anticipate appointments where hemoglobin A1c will be measured in quasi-periodic fashion (right). Similarly, Figure 6 (left) illustrates the ability to model the rates of complications. Medical guidelines do not specify scheduling for regular follow-up of the adverse events, and this is congruent with the lack of periodicity in the KNR hazard predictions.

**Forecasting, multi-forecasting, and interrogability.** As one would expect, a trade-off occurs between early prediction and predictive performance. The effect of WRN-PPL forecast distance $c$ on KNR prediction is shown in Figure 6 (right). Notably, the 3-month censored WRN-PPL has approximately the same performance as the nowcasting LSTM[2]. Similarly, effects of single-model, multiple-$c$ prediction are shown in Figure 7 (left), illustrating the WRN-PPL improvement over WRN for nowcasting (up to $c < 1$) but not for $c \geq 1$. The coefficient profile as a function of $c$ in multiple-$c$ prediction is also shown (middle left), demonstrating that relative-time attributions, which are commonly used in association statements in health literature, appear to depend on censor time $c$.

Figure 7 (middle right and right) shows the wavelet reconstruction for the effect of troponin level and timing on rate. Both reconstructions demonstrate recovery that acute coronary syndrome is diagnosed within the next time unit after a troponin greater than 0.01 ng/mL. The multiple-$c$ reconstruction on the right more accurately reflects the uniform distribution hazard, namely, increasing hazard if the event has not yet occurred.

## 6 DISCUSSION

The performance of WRN-PPL in Table 1 and Figure 5 illustrates the utility of our model, in particular in identifying the near-periodicity of recurring events. For example, the rate prediction for the individual denoted in green in Figure 5 (right) suggests that individual may have skipped, missed,

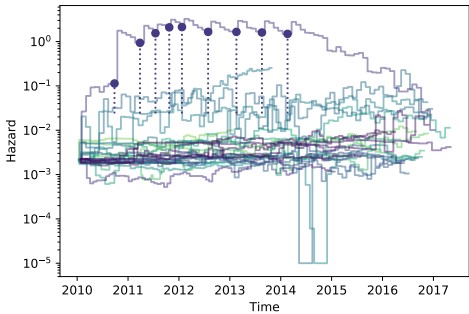 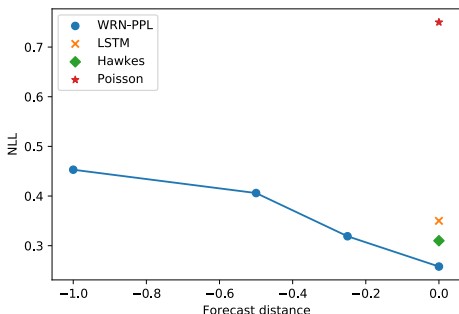

Figure 6: Left: ketoacidosis, retinopathy, or polyneuropathy diagnoses and rate predictions for thirty random test set patients. Right: combined outcome (KNR) negative log likelihood as a function of censor distance (in years).

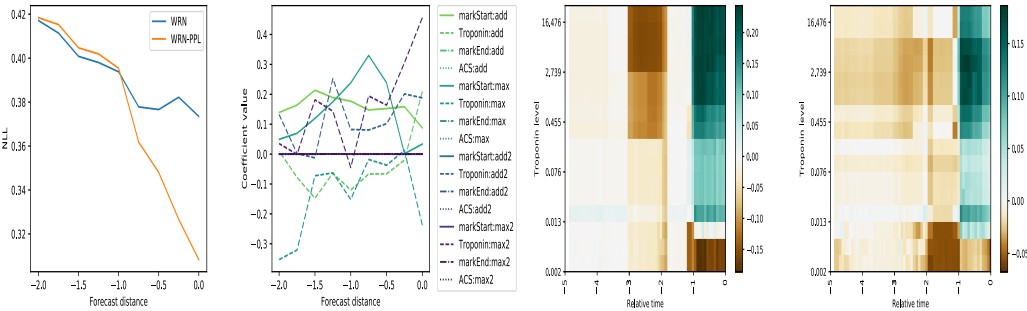

Figure 7: Best viewed in color. Left: negative log likelihood as a function of forecast censoring distance $c$ for multi-forecasts. The permute and pool layer expresses greater expressivity to model the hazard than WRN. Middle left: coefficient profile as a function of forecast censoring distance $c$ for WRN. The reduction layer comprises additions and maximums of 1-d and row-indexed 2-d reconstructions. Middle right: wavelet reconstruction of troponin contribution to ACS hazard. A preceding troponin above 0.01 indicates increased rate of ACS occurrence within the next hour. Right: WRN-PPL reconstruction image for multi-forecasting at $c = \{0, 0.25, 0.5, 0.75, 1\}$.

or rescheduled 5 to 6 appointments over the last decade. The peaks reach hazards of approximately 3, indicative of a mixture of belief and uncertainty–belief that in those months the event should occur at a rate above three per year, and uncertainty about the occurrence of the appointment.

For multi-forecast learning, a comparison of the results in Table 1 and Figure 7 (left) demonstrates the value of model expressivity. In particular, Table 1 shows that single forecasting outperforms multi-forecasting at $c = 0$ in Figure 7 (right). However, Figure 7 (middle right and right) illustrates that multi-forecasting improves the learned wavelet representation. These findings suggest that the layering between the wavelet reconstruction (WRN: {reduction layer, linear}, and WRN-PPL: {reduction, permute and pool, linear}) and the hazard output is not adequately expressive to map the true wavelet reconstruction to the true hazard. We argue the solution is not in simplification nor abandonment of the multi-forecast setting, but in leveraging the multi-forecast setting to facilitate recovery of the wavelet reconstruction by using an even more expressive mapping.

**Conclusion.** Wavelet reconstruction networks is a forecasting method tailored for health care settings. Its advantages include multi-resolution representation of relative time dependencies in 1 and 2 dimensions and enables time step specification at test time. The combinations of relative time functions result in a model with powerful predictive capacity, flexibility, and interrogability. We demonstrated improved performance of our system over competing methods in a analysis of diabetes and showed the ability to capture quasi-periodic events that could be used to measure adherence and forecast risk of complications.

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

## APPENDIX A    ADDITIONAL EXPERIMENTAL DETAILS

### A.1    PARAMETER SEARCH AND SETTINGS

There are number of settings to provide for the implementation. We set the number of absolute time steps to 80, equal in size. We treat categorical variables both as empty marks and as an indicator variable per category. We use one- and two-dimensional Haar wavelets with 64 relative time bins and 16 value bins. The time bins are linearly spaced from 0 to the largest relative difference of event and target in the training set. If a feature's train set value distribution is both positive and negative, the bins are mapped linearly across the value range, and if values are of one sign, bins are spaced linearly in the space of a $\log(1 + |x|)$ feature transformation. In our experiments we use a single permute-pool layer with $P = 4$. We use the Adam optimizer with manual setting of the learning rate $10^{-3}$, event/reduction regularization parameter $\gamma_\beta = 10^{-7}$, and wavelet regularization parameter $\gamma_w = 1/N$, all based on tune set performance. We use a mini-batch of size 10. For ease of computation, we randomly subsample features whose number of occurrences exceeded 50 to 50 samples. Inspection of the data suggests that the subsampling would have negligible effect on performance while reducing memory requirements substantially. Dates of events were perturbed across years and subsampled for further anonymity. We penalize hazards below $10^{-5}$ by a large constant 100 per unit time on the train set, and pass all predictions through a rectified linear unit to ensure valid hazards. We set a max hazard gain of $\log(10)$ to ensure numeric stability. These constraints are appropriate for our tasks, as rates below $10^{-5}$ and above 10 are not meaningful for our applications. For LSTMs, we set $h = 4$ and 16 for the simulated and real data experiments. We searched over $\{4, 8, 16, 32, 64, 128\}$ LSTM hidden unit sizes, optimizer learning rates of $10^{-2}, 10^{-3}, 10^{-4}$, and step sizes $\{40, 80, 160\}$ using only the train and tune sets. We set event/reduction regularization parameter $\gamma_\beta = 10^{-7}$ and wavelet regularization parameter $\gamma_w = 1/N$, LSTM[1] hidden sizes are 128, and LSTM[2] hidden sizes $h = 4$ and 16 for the simulated and real data experiments respectively based on tune set performance.

### A.2    DESCRIPTIVE STATISTICS

Table 3 provides basic information about demographics and the outcomes for the regional cohort patient population after inclusion and exclusion criteria have been applied.

Table 3: Descriptive statistics of the diabetes study population, reported with median [2.5%,97.5%] and $n$ (fraction).

| Feature | $n$=4,732 |
|---|---|
| Age in 2010 | 35 [0, 82] |
| Gender | |
|    Female | 2526 (0.53) |
|    Male | 2204 (0.47) |
| Type I DM | 45 (0.01) |
| Type II DM | 459 (0.10) |
| HbA1c | 980 (0.21) |
|    First | 5.9 [4.8,10.2] |
|    Any | 6.7 [5.1,10.4] |
| Combined | 137 (0.03) |
|    Ketoacidosis | 15 (0.00) |
|    Polyneuropathy | 88 (0.02) |
|    Retinoptahy | 34 (0.01) |

## APPENDIX B    HAZARD EXAMPLES

Additional test set patient hazard predictions are given alongside occurring events. The figure shows 5 random test examples for hemoglobin A1c using WRN-PPL for nowcasting. Calibration plots suggest the hazard predictions are accurate for high hazards, and 1.5- to 4-fold overestimates for low hazards (though the absolute error is small).

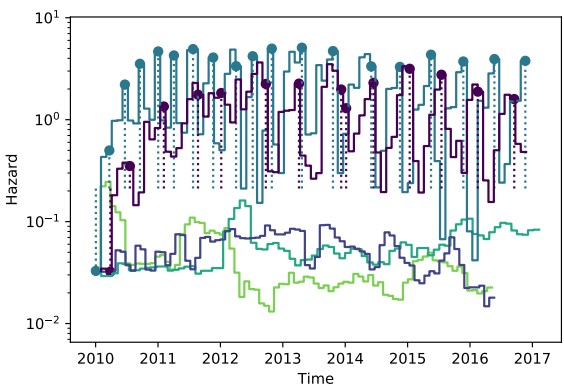

Figure 8: Hemoglobin A1c hazard prediction with nowcasting

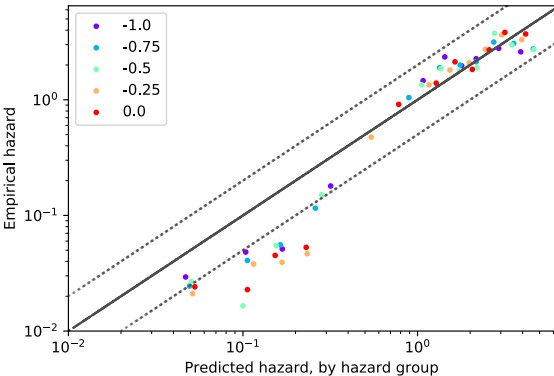

Figure 9: Hemoglobin A1c calibration plot for multi-forecasting, with $c = \{0, 0.25, 0.5, 0.75, 1\}$. The legend denotes the number of years prior the time at prediction is compared to the forecast time. Dotted line denote 2-fold miscalibration.

## APPENDIX C  LOSS-BY-EPOCH CURVES

Epoch curves were used for early stopping–the parameters used on the held out test set were those for which the tune set negative log likelihood was minimized, only for the best performing hyperparameter setting (identified through tune set performance).

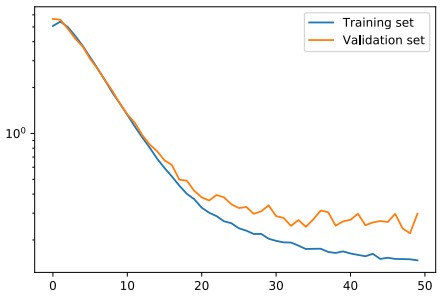
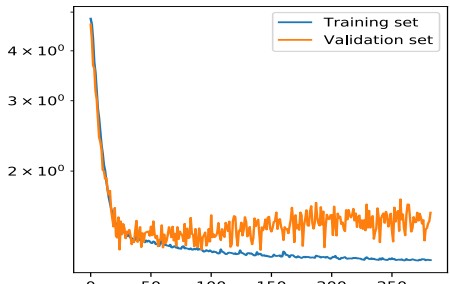

Figure 10: Loss by epoch curves for real data, target: {ketoacidosis, polyneuropathy, retinopathy} used for early stopping (left) and target: hemoglobin A1c (right).

