# OpenReview forum: "Clinical Risk: wavelet reconstruction networks for marked point processes"
_ICLR.cc/2019/Conference_

### Official Review · AnonReviewer3 · 2018-10-24
**Learning shallow Hawkes kernels using wavelets**

**Rating:** 5
**Confidence:** 4

**Review:**

This paper centers around efficient estimation of the kernel function for the Hawkes process and relaxation of the “linearity” assumption in the original Hawkes process. They rely on a classical sparse generalized linear model using the wavelet basis set and Hawkes loss function to estimate a shallow kernel function. This approach is opposite to the deep function estimation approach which does not rely on a predefined basis set [e.g. see [Du et al, 2016]]. However, it can have an advantage that the learned functions are interpretable, thought the authors never demonstrate it in the paper.

Given this view of WRNs (an unfortunate coincidence with WideResNets), we understand how LSTM2 outperforms LSTM1 in the results. However, the results tables do have peculiar numbers too. For example, why the Goodman-Kruskal gammas for H. Poisson are exactly -1? Why is it always pointing in the wrong direction? There are other observations in the results table that the authors have listed without much explanation. For example, in Section 5, what is the reason for “The WRN-PPL method excelled particularly in tasks with many target occurrences”?

Another example is the arguments in the discussion section about the use-case of rate functions. For example, the authors state: “ For example, the rate prediction for the individual denoted in green in Figure 5 (right) suggests that individual may have skipped, missed, or rescheduled 5 to 6 appointments over the last decade.” How did the authors conclude this claim? What is the clinical significance of missing or rescheduling 5-6 appointments in the context of A1c prediction?

Writing can be seriously improved (basically the paper is not ready in the current state). For example, only in Section 6, the authors have introduced the full name of WRN-PPL after using it many times before.

The motivation for this paper is misleading. There have been several works on “Deep Cox” and “Deep Hawkes” models. I don’t see the novelty in the authors’ contribution in defining the clinical risk. Especially Fig. 3 (left) is already known and does not add much value.

Overall, on the positive side, this paper shows that in some datasets going back to the classical shallow models we might achieve better performance than the alternative deep models. Unfortunately, the authors do not clearly state how many training data points they have. They have a vague statement: “798,818 timestamped events in a study population of 4,732 individuals”, but it does not say exactly how many training examples they have.

---

> ### Author Response · Authors · 2018-11-18
> **Thank you for the feedback.**
>
> '''learned functions are interpretable, thought the authors never demonstrate it'''
> Interrogability is illustrated in Figure 7, where the timing and event value interaction is shown to affect the rate of ACS calls.  Additionally, WRN has GLM form, and common interpretations akin to linear, logistic, and Cox regression can be made.  GLM interpretations are challenging when the number of features is large, and this is also the case for WRN.
>
> ''' For example, why the Goodman-Kruskal gammas for H. Poisson are exactly -1? '''
> H. Poisson uses the training set average rate for all examples. Thus every example has the same predicted rate and the same average empirical rate:  (count / period).  Every pair is a tie, and ties are dropped, so only the prior terms remain and the result is -1 (the estimate of the statistic is undefined if prior = 0).
>
> ''' ... without much explanation.  For example, in Section 5, what is the reason for “The WRN-PPL method excelled particularly in tasks with many target occurrences”? '''
> When the number of target events is larger (A1c), complex models with more parameters (WRN-PPL) are likely to perform better.
>
> ''' For example, the rate prediction for the individual denoted in green in Figure 5 (right) suggests that individual may have skipped, missed, or rescheduled 5 to 6 appointments over the last decade.” How did the authors conclude this claim? What is the clinical significance of missing or rescheduling 5-6 appointments in the context of A1c prediction? '''
> Standard US medical guidelines require A1c measurement at least as frequent as 6 months for those with diabetes. One of the authors (a physician) compared this recommendation with the A1c measurement timestamps and concluded 5-6 appointments were missed.  Appointment adherence is clinically important as clinical decision making about treatment regimen, eligibility for e.g. bariatric surgery, depend on a patient's adherence to routine engagement with health care.
>
> ''' Writing can be ... improved '''.
> We thoroughly revised section 3 for notational clarity.  WRN-PPL abbreviation is now included there.
>
> ''' There have been several works on Deep Cox and Deep Hawkes ... '''
> Deep Cox and Deep Hawkes, are only evaluated on the t'=t_0 line and the t'=t lines respectively.  There is a half-quadrant of Figure 3 pertaining to prediction tasks unaddressed by these works.
>
> ''' Training data points ... vague '''
> 4,732 individuals corresponds to 4,732 trajectories (1 trajectory per individual).  Training examples is 1/3 of that, rounded up: 1,578.  Revised in text.

---

> > ### Comment · AnonReviewer3 · 2018-11-19
> > **Thank you, but still confused**
> >
> > Thank you for updating the draft. Here are my comments on your response.
> >
> > I see your point now. Figure 7 is not very readable for me though.
> >
> > I think H. Poisson is not a meaningful baseline to include based on your description.
> >
> > If the reason for better performance is the higher number of parameters in your model, then the comparison is not fair. Basically, the baseline models are underfitting.
> >
> > I think the insight provided by the algorithm is not very useful. You can easily look at the patient records and see if he has missed doctor appointments; you don't need a learning algorithm for that. Moreover, the treatment adherence measurement and control is a well-studied field in behavioral science and you need to provide stronger pieces of evidence that this algorithm can play any role in it. The current explanation is only hand-waving.
> >
> > Regarding deep Cox and Hawkes models:
> > I am still not clear what is the novelty of your contribution. The goal of all rate estimation techniques such as Cox and Hawkes models is to estimate a rate \lambda(t, X), where t is an arbitrary time in future and X represents a slice of history of the patient. While the very original definitions of Cox and Hawkes models had different assumptions on the timing of the prediction and slice of the patient history that are used, the numerous extensions have covered different options for these two. Basically, the difference between these two only boils down to the loss function. For example, this paper is in the category of generalization of the Hawkes process because of Eq. (1). People have also used the relative time idea before too. So, I am not sure why the authors claim any novel contribution in this discussion, it should be left as a discussion and cite the previous papers that have studied rate function estimation under relaxed prediction and forecasting times.

---

> > > ### Author Response · Authors · 2018-11-19
> > > **Addressing your comments**
> > >
> > > Thank you for your comments.  We will work to integrate them into the manuscript.
> > >
> > > '''H. Poisson is not a meaningful baseline to include based on your description'''
> > > Poisson rates are commonly used, i.e. whenever someone models events in aggregate as counts over periods. We feel it is important to know how much better we are doing than the simple baseline.
> > >
> > > '''You can easily look at the patient records and see if he has missed doctor appointments.'''
> > > It turns out this is a hard problem and there is a whole literature on appointment adherence.  The EHR provides a history of scheduled and completed appointments, but rescheduling is common.  Using only the completed appointments is also problematic because there are visits for all sorts of maladies, not ones that are focusing on diabetes care.  A1c is a good proxy for having a visit with respect to diabetes (it is highly specific to outpatient diabetes care).  Predicting visits for diabetes based on doctors appointments is problematic because sometimes a patient goes to a family practitioner, an internist, an endocrinologist, and who they see is not at all standard.  Translating the risk function into an alert system is ongoing but we believe it is outside the scope of this work.  But, briefly, you want to know how early you can make an alert because early is better (more time to perform intervention), and unless you do multi-forecasting you don't have a sense of the tradeoff in prediction quality and early alerting.
> > >
> > > '''Higher number of parameters in your model, then the comparison is not fair'''
> > > We respectfully disagree. It is common to compare MLPs with logistic regression, decision trees, ensembles, etc.  The number of parameters differs significantly across these methods and does not invalidate the comparison. In our experiment, the LSTMs have far more parameters than our model, and we outperform them.  The Hawkes model we compared against has fewer parameters, and we also outperform them.  We used the tune set and early stopping to select hyperparameters for the various methods to prevent overfitting, but this did not completely resolve the implicit bias-variance tradeoff.
> > >
> > > '''Regarding deep Cox and Hawkes models: ...'''
> > > We are aware of Deep Cox models, e.g. including two recent publications, DeepSurv (Katzman et al 2018) and DeepHit (Lee et al 2018), which are evaluated on vertical lines of Figure 3.  We are not aware of any Deep Hawkes processes that move from nowcasting to forecasting and multi-forecasting.  Multi-forecasting is made explicit by the visualization in Figure 3, and we are not aware of such a figure in prior art.

---

> > > > ### Comment · AnonReviewer3 · 2018-11-20
> > > > **Response and a request**
> > > >
> > > > Regarding H. Poisson: I disagree, but this is a minor issue compared to the rest.
> > > >
> > > > Regarding the usefulness:
> > > > 1) Given the literature in behavioral studies and treatment adherence, your discussions are still hand-waving and far from being rigorous.
> > > > 2) The fact the in the U.S. there is not a unified records for patient history does not mean that we should use an algorithm to infer that. That is not a right way to solve it. However, recently Apple has started to enter this domain to provide a centralized access to the records. There are also a couple of insurance companies such as BCBS who are trying to accomplish this task. When the right solution is just a look up, you don't use a "Wavelet Hawkes" model to do it.
> > > >
> > > > Regarding number of parameters:
> > > > I disagree and leave it to the AC to decide about this. In choosing the baselines, you need to choose a reasonably complex from common class of models and make sure that they don't underfit. Basically, the baselines are too weak.
> > > >
> > > > Regarding multi-forcasting:
> > > > I have a request: can you adjust the notations in the introduction (or write a new dedicated paragraph to this) and use t' in the descriptions as you have used in Fig 3 (right)? Then I will provide more discussions and references.

---

> > > > > ### Author Response · Authors · 2018-11-20
> > > > > **Thanks and revised.**
> > > > >
> > > > > Usefulness:
> > > > > Appointment adherence and treatment adherence are separate areas, and we maintain that predictive models forecasting A1c as a measure of diabetes appointment adherence is useful and central in clinical decision making.   We argue that forecasting adherence is one of the use cases where a predictive model can more readily be used--because risk to individuals is minimal if predictions are wrong (i.e. less ability to detect maladherence), and because an existing "look-up", while ideal (for retrospection at least), is as you say, not available.
> > > > >
> > > > > Parameters:
> > > > > The comparison methods are expected to outperform the models in common use in health care, e.g. the RECODe equations (Basu 2017, 2018) would not work well because we have recurring events.  We also compare against a leading healthcare LSTM (Lipton et al 2016) and the closest LSTM model.
> > > > >
> > > > > Multi-forecasting:
> > > > > Thanks, revised in text.

---

> > > > > > ### Comment · AnonReviewer3 · 2018-11-22
> > > > > > **Thanks, this is clearer**
> > > > > >
> > > > > > @Usefulness and parameters:
> > > > > > There is no point in back and forth argument. I have provided my argument for these two and AC can read and decide whose argument is more convincing.
> > > > > >
> > > > > > @Multi-forecasting:
> > > > > > Now I better understand what you have done in multi-forecasting. Based on the descriptions after Eq. (3), I see the multi-forecasting trick (using multiple c during training) as a data augmentation/dataset expansion tool. This makes total sense to me. But I don't see much novelty in Figure 3 (right), but this is ok.
> > > > > >
> > > > > > Recommendation:
> > > > > > I suggest to define the rate function as \lambda(t, t', x_{<= t_0}), where:
> > > > > > t: forecast time, the time we want to know the hazard.
> > > > > > t': the prediction time,
> > > > > > t_0: study time.
> > > > > >
> > > > > > Nowcasting: t_0 <- t' (which is slightly more meaningful that t' <- t_0). I think this is a clearer notation.

---

> > > > > > > ### Author Response · Authors · 2018-12-01
> > > > > > > **Re: recs**
> > > > > > >
> > > > > > > Thanks--we have modified the paragraph's notation for added clarity and will be reflected in the next version.

---

### Official Review · AnonReviewer1 · 2018-10-31
**Well-motivated and innovative approach to construct intensity functions using wavelets. But the overall quality is not good enough due to large amount of unclear important content (in both method and experiments).**

**Rating:** 4
**Confidence:** 4

**Review:**


[PROS]

[originality]

The paper proposed to construct the intensity function of a point process using wavelet, in order to improve its expressiveness, e.g. allowing non-additivity.

The authors did extensive experiments to investigate their model performance compared to many appropriate baselines, on both synthetic and real-world datasets.

[CONS]

[clarity]

The major drawback of this submission is its clarity. The paper is vague at various important points in both method and experiments, thus leaving their correctness and soundness undetermined.

In the method section, the authors did not specify a well-defined and self-consistent notation system. This makes the paper really hard to understand. For example, one may be easily confused with things like:
1) How q, q(g(t), t_i), g(t), g_{es}, g_{d} are connected and distinguished?
2) The function g_{es} maps from t to R, then is t a space or a variable?
3) The state s seems crucial in the function g_{es}, but why it is only mentioned one time in the paper? How it is defined and how it is used?
4) How the Hadamard product is applied to two matrices of different sizes?
5) j=1 is time dimension and j=2 is time and value dimension, then why j=1 is not part of j=2? Time is needed in both cases and it seems natural that the associated parameters are shared.
6) Figure-3 has t_i in figure, t’ in caption, but the text in main paper mentions t_0 for Figure-3. How are they related? Are they actually the same?

The most confusing part is the censoring distance c. Its introduction around eqn-(2) suggests that c > 0 and the censoring section clearly mentions that. But c is also used to denote the forecast distance, which is clearly < 0 according to Figure-6 and Figure-7. What’s worse, there is also a term called forecast censoring distance. What are the relationships among these terms? If they are all the same, then is the c actually a model parameter or an evaluation control knob? Such things are very important to clarify.

Moreover, the paper did not clearly explain how the model is trained in each case, especially for (multi-)forecasting. In details:
1) What is the training objective?.
2) What is the optimization method for this objective?
3) How is it implemented and would the code be released?

It is good that the experimental section lists many appropriate baseline models and multiple evaluation metrics, but it is not clear how they are used. For example:
1) Fourier methods and Hawkes process do not deal with the value v, then how are they fairly compared to the proposed model which takes v into account as in eqn-(2)?
2) How is the Goodman-Kruskal gamma exactly computed? On all the instances of the held-out set? What is exactly the rank in this case?
3) The authors also leave out the positiveness constraints of a Hawkes process to incorporate inhibitory interactions, but how the positivity of the intensity function is ensured in this case?

[quality and significance]

The method is well-motivated and innovative. But details of the model and experiments are very unclear, so its overall soundness is hard to judge. For example:
1) The authors claim that, compared to neural models, their model has the advantage of the interpretability (for small datasets), but they also have neural components in their model. So why their model is more interpretable than others (e.g. Mei and Eisner 2017 as they cited) is not clear to me.
2) It is not clear why the interpretability is associated with the size of the dataset (quote `remains interpretable for small data sets’). What’s worse, the interpretability seems the only advantage of the model over other neural models (please correct me if I am wrong). If this edge could not scale up to large datasets, then does it mean on large datasets, a neural model should always be preferred over this model, because they are supposed to be more expressive?

---

> ### Author Response · Authors · 2018-11-18
> **Reorganization and reformulation of absolute- and relative-time kernels**
>
> Thank you for the detailed comments about the notational clarity.  With your aid we have reorganized section 3, specifically defining absolute-time kernels (h) and presenting the rate function as a linear combination of reductions of them.  We: (1) state the rate function form, (2) define h, and (3) specify the causally-protective translation function from g to h.
>
> Methods:
> - The notations regarding the translation function has been fixed and updated in the revised PDF.
> - Mapping g_es is fixed. s is defined further in the revised text as a value interval.
> - The Hadamard product example is modified to include a column vector of 1's denoting dimension expansion.
> - The j=1 is the temporal analogue of a missingness indicator in atemporal analysis.  That the event occurred may be independently informative of the value of the event in the same way as a missingness indicator can improve prediction. Thus it makes sense to include both kernels for j=1 and j=2.
> - We have fixed Figure 3, the two time dimensions figure in alignment with the text.  Namely, t is the (absolute-time) forecast time, and t' is the (absolute-time) time at prediction. The time t' is the placeholder for the time dimension corresponding to the t_i which is the time-stamp of observed event e_i.
> - c is a non-negative value, however, it is visually more common to view time from left to right.  We will denote that the axes correspond to -c values.
>
> The multi-forecast setting is now stated (as the average negative log likelihood across as vector of pre-chosen c).  Optimizer details are in the attached Appendix.  Code will be released upon publication.
>
> Experiments:
> - The Fourier methods do include the values of the events using the same value intervals (s \in S).  The Hawkes process does not, and the performance hit they take is shown in the ACS example where the numeric value is central to prediction. Revised in text.
> - Goodman Kruskal statistic is computed using all test set pairs.  For every test set pair with different counts of outcome events, compare the empirical and predicted average hazards (count/period and cumulative hazard/period) of each.  If the difference in counts has the same sign as the difference in predicted cumulative hazards, this increases the "agree" count, otherwise this increases the "disagree" count.  The formula is (# agree - # disagree - prior) / (# agree + # disagree + prior), where we chose prior = 1000.
> - We require a positivity constraint as negative rates are not well defined.  Revised in text.
>
> Interrogability:
> - Figure 7 illustrates the interrogability of the model, as the relative effect of a kernel on the rate can be visualized. In the ACS example, it is evident the troponin, only at specific relative times and values, impacts the call of ACS.
> - Furthermore, WRN has GLM form, which are commonly used for relationship estimation (linear, logistic, Cox) and this model allows for those types of interpretations.  GLMs suffer in interpretability when 100+ features are used, and the same is true for this model.  Large sample sizes do not impact interpretability of our model beyond the standard wider confidence intervals in parameter estimates.
> - Mei and Eisner 2017 does not allow for marked events, is limited to nowcasting, and does not have GLM form.  Revised in text.

---

> > ### Comment · AnonReviewer1 · 2018-11-29
> > **Need further clarification**
> >
> >
> > Thanks for your clarification in both rebuttal and paper.
> >
> > > The Hawkes process does not, and the performance hit they take is shown in the ACS example where the numeric value is central to prediction.
> >
> > Table-1 shows NLL of Hawkes process on your data. If the data has any mark the model does not handle, then the NLL would be \infty. Why not so in your case? How do you make sure your comparison is fair in this case?
> >
> > > We require a positivity constraint as negative rates are not well defined.
> >
> > The Mei and Eisner 2017 you cited designed such a model called D-SM-MPP. What is the difference from that? If no crucial difference, maybe you should note that in your paper.
> >
> > > Figure 7 illustrates the interrogability of the model, as the relative effect of a kernel on the rate can be visualized.
> >
> > Can you clearly distinguish interrogability and interpretability? You mention both in your paper but did not clarify. If there is no difference, why not use interpretability which seems more commonly known?
> >
> > Moreover, since interrogability is a major contribution that you claim, do you have a way to evaluate its accuracy? (Not only showing the figure but also somehow decide if your figure makes sense.)
> >
> > > Large sample sizes do not impact interpretability of our model
> >
> > Please explain your sentence *our method remains interpretable for small data sets*
> >
> > > Mei and Eisner 2017 does not allow for marked events, is limited to nowcasting, and does not have GLM form.
> >
> > Neither does Hawkes process. But you compare to Hawkes process anyway. Can you clarify?

---

> > > ### Author Response · Authors · 2018-12-01
> > > **Thanks, some details**
> > >
> > > """If the data has any mark the model does not handle, then the NLL would be \infty. Why not so in your case? How do you make sure your comparison is fair in this case?"""
> > > The outcomes of interest are not marked, but the features may be.  Thus, if our features processing e.g. in Hawkes captures the timing but not the mark, there is no contribution of marks to the hazard, but the hazard remainds finite.  W.r.t. fairness, our goal is to compare against (1) closely related classic baselines and (2) closest recent methods.  Hawkes is a closesly related classic baseline and (Lipton et al) LSTM1 and (our preprocess) LSTM2 are closely recent methods because they are high performing health care deep learning temporal models, and they account for marks.
> > >
> > > """The Mei and Eisner 2017 you cited designed such a model called D-SM-MPP. What is the difference from that? If no crucial difference, maybe you should note that in your paper."""
> > > Thanks, we use the dual form of the constrained optimization requiring lambda >= 0, applying a large and linearly-increasing penalty factor for negative values.  This is described in the Appendix. D-SM-SPP is a generalized Hawkes process (Daley 1988) with the activation defined as softplus.  We were concerned that the softplus approach might have adverse gradient step update effects since hazards are often close to zero, mandating very large negative value predictions as inputs into the softplus and subsequent instability of the backprop gradient calculation.
> > >
> > > '''Can you clearly distinguish interrogability and interpretability? You mention both in your paper but did not clarify. If there is no difference, why not use interpretability which seems more commonly known? Moreover, since interrogability is a major contribution that you claim, do you have a way to evaluate its accuracy? (Not only showing the figure but also somehow decide if your figure makes sense.) '''
> > > - Within the interpretability literature, there is a push for recognition that interpretability is the high level descriptor (i.e. somewhat vague), and that other phrases should be used to state the characteristic of interpretability the model provides.  We believe that the visualizations admitted by our model provides interrogability.
> > > - Interpretability often requires usability studies, e.g. surveys, and quantified downstream evaluation.  We felt this was out of scope for this submission.
> > >
> > > """Please explain your sentence *our method remains interpretable for small data sets*"""
> > > Thanks, revised in text (the next version) to denote interpretability for data sets with few features.
> > >
> > > """ > Mei and Eisner 2017 does not allow for marked events, is limited to nowcasting, and does not have GLM form. Neither does Hawkes process. But you compare to Hawkes process anyway. Can you clarify?"""
> > > - Please see the first response in this comment for our reasoning about choice of comparisons.

---

### Official Review · AnonReviewer2 · 2018-11-05
**It introduces wavelet reconstruction to the temporal point process literature**

**Rating:** 7
**Confidence:** 4

**Review:**

The authors of this paper propose a point process model that uses wavelet reconstruction to capture complicated dependencies between events. They motivate this approach using experiments in the medical domain, which show how certain dependencies between events is better captured by their proposed model. The primary contribution of the paper is novel and could be useful in medical settings where predicting occurrence of important events such heart attacks could be challenging with alternative methods.

A few recommendation for improving the paper besides a few typos that are present in the manuscript (such as first sentence of section 3 where  "depicted" is redundant): 1) A more substantive discussion of the challenges one may encounter while training the proposed model and elaborate on the complexity of the inference procedure in comparison with alternatives. For example, training vanilla Hawkes model is relatively easy and efficient, even though it may not perform as good as the proposed model. 2) Presentation of the model in section 3 lacks sufficient explanation and heavily relies on high level remarks on how the model is developed, a more detailed explanation (even including how the wavelet coefficients are computed) could be far more useful than the schematic view of the network architecture presented in Figure 1.

Overall, I think this paper provides a novel contribution for modeling event data specifically for medical data. The ideas are well presented and experiments provide insights in how the proposed model can be useful for forecasting particular medical events. The overall recommendation is to accept the paper, however, I hope authors would address the concerns provided in the previous paragraph.

---

> ### Author Response · Authors · 2018-11-18
> **Thanks, and expansions**
>
> Thanks for your comments.  We agree that this model will effectively model marked, time-stamped medical data, and it does so with fewer limitations from existing competitor methods (e.g. LSTMs and time step issues, e.g. neural Hawkes processes and lack of marked events and limited to nowcasting).
>
> (1) One limitation not mentioned is the following: when the frequencies of events differ by many orders of magnitude (e.g. pulse oximiter measuring every second intermixed with colonoscopy once every 5 years), we have to make a choice to initiate a signal with every event or subsample the high frequency events.  We subsample for efficiency but using high frequency events as a single signal instead of many signals may be more appropriate.  The size complexity is shown in the architecture plate diagram where the complexity is given by the indices of the plate, and optimization details are included in the Appendix previously omitted.
>
> (2) We reorganized Section 3 on Wavelet Reconstruction Networks to make model details precise.  This includes the definition of kernel functions on both absolute time (h) and relative time (g) as well as the translation problem when using discretized time intervals.

---

### Meta-Review · Area_Chair1 · 2018-12-14
**Meta-Review for Clinical Risk paper**

**Confidence:** 3
**Recommendation:** Reject

**Metareview:**

There was discussion of this paper, and the accept reviewer was not willing to argue for acceptance of this paper, while the reject reviewers, specifically pointing to the clarity of the work, argued for rejection. There appear to be many good ideas related to wavelets, and hopefully the authors can work on polishing the paper and resubmitting.